**Comment**

# Reframing chemistry education in the age of automation and AI

Dirk G. Kurth

Chemistry programmes worldwide are experiencing declining student enrolment, reflecting shifting career perceptions and competition from emerging fields. This comment outlines how the discipline is evolving into an integrative field that combines molecular science with data science, automation, and systems thinking, thereby necessitating an evolution of both research practices and educational structures.

Studying chemistry has always demanded perseverance, focus, diligence, and a considerable degree of frustration tolerance. Laboratory work can be exhausting and requires stern discipline. Traditional chemistry curricula place a strong emphasis on practical training in synthetic and physical chemistry. Synthetic chemistry, in particular, remains a craft whose mastery requires years of practice, whether working in a fume hood, handling a Schlenk line, or manipulating microgram quantities of reagents. Likewise, analytical and physical chemistry rely on a deep understanding of mathematics, physics, and quantum mechanics. In short, choosing chemistry is a serious investment.

For decades, this investment was rewarded. Chemistry promised stable employment and above-average salaries, even compared to competitive fields such as automotive, engineering, pharmaceuticals, or medical technology. Yet the question now looming over prospective students, young academics, and Universities is whether this promise still holds.

## A decline in enrolments - a shift in perception

Over the past decades, most of Europe, the United States, and parts of Asia have seen a persistent decline in undergraduate chemistry enrolments. Nobel laureate Harry Kroto[1] warned as early as 2004 about the troubling drop in undergraduate chemists in the UK. The causes are complex. The public associates chemistry with per- and polyfluoroalkyl substances (PFAS or so-called forever chemicals), microplastics, and pollution more readily than with battery materials, photovoltaic films, pharmaceuticals, or smartphones. Media narratives foreground environmental damage and litigation, while chemistry's central contributions to energy, health and technology receive comparatively little attention.

This decline in interest is amplified by labour-market perceptions. The chemical industry, once associated with stability and long-term career prospects, is undergoing a structural transformation driven by decarbonisation, electrification, geopolitical tensions, and supply chain reconfigurations. While its societal relevance remains substantial, career pathways appear less transparent to prospective students than in rapidly expanding fields such as information technology or engineering. Importantly, these perceptions are not independent of education. Students interpret the

relevance of a discipline through its training structures and its visible connections to emerging technologies. When chemistry curricula do not clearly reflect contemporary developments, such as automation, AI-driven discovery, and interdisciplinary integration, the discipline risks being perceived as less aligned with future-oriented careers, irrespective of its actual strategic importance.

**Rising competition from other fields.** Students capable of succeeding in chemistry would likely perform well in many other demanding programmes. In Germany and elsewhere, young people increasingly gravitate toward business, computer science, and electrical or mechanical engineering, which are perceived as offering clearer career trajectories, higher flexibility, and stronger alignment with emerging technological sectors. Indeed, apart from traditional science programmes, the total number of students at German Universities has increased over the past 20 years. In particular, Universities of Applied Science, with their focus on industry-oriented degree programmes, have greatly benefited from this trend[2]. At the same time, chemistry graduates possess highly transferable skills, including analytical thinking, quantitative reasoning, and problem-solving, which enable them to transition into a wide range of professions beyond traditional chemical research. While this flexibility is a strength, it also highlights a structural issue: the pathways from chemistry programmes to clearly defined professional roles are often less visible than in competing disciplines.

Students evaluate fields of study based on their perceived alignment with contemporary science and career prospects. When chemistry curricula emphasize traditional laboratory training but underrepresent AI-driven methods, automation, and interdisciplinary applications, the discipline risks appearing less connected to current scientific practice and industrial needs. Addressing this perception gap is, therefore, not only a matter of communication but of aligning education with the evolving reality of chemistry.

**Chemistry, electronics, and AI: A deep interdependence.** Despite declining student numbers, chemistry's strategic importance has not diminished, quite the opposite. Since the 1950s, chemistry and electronics have been the two most transformative technological sectors. The electronics industry would simply not exist without chemistry: polymeric insulation, advanced dielectrics, etchants, high-purity solvents, thin-film materials, semiconductor precursors, and sophisticated photoresists form the foundation of microfabrication, to name but a few. There would be no batteries, windmills, photovoltaics, or smartphones without chemistry[3].

Meanwhile, electronics and computing have transformed chemistry. Fourier-transform spectroscopy, imaging, quantum chemical modeling, and high-performance computing have become integral to modern chemical research. AI-driven discovery recently tackled one of biology's greatest challenges, predicting protein structures, which earned the 2024 Nobel Prize in Chemistry[4]. Notably, this breakthrough originated at a tech company, DeepMind, rather than in a traditional chemistry department. At the same

time, the Nobel Prize in Physics was awarded to two computer scientists who laid the foundational concepts of modern AI[5]. These awards underscore two points: first, AI is becoming a central tool in chemical and materials research, and second, AI bridges traditional disciplinary boundaries.

**A research landscape being transformed.** The global push toward sustainable and circular chemical processes imposes Herculean research demands. Many experimental workflows, such as catalyst discovery, polymer library development, battery materials screening, and reaction optimization, are slow, labour-intensive, and prone to irreproducibility[6]. Automation, robotics, AI-driven discovery, and high-throughput experimentation are now reshaping the landscape[7].

Self-driving laboratories, integrating AI with autonomous robotic platforms, can design, perform, and analyse experiments with minimal human intervention. AI-based models predict molecular properties, reaction pathways, and optimal experimental sequences[8]. Automated reaction platforms, microfluidics, and integrated analytical characterization make it possible to explore chemical and materials spaces orders of magnitude faster than manual experimentation ever allowed[9]. The shift toward automated discovery is not merely convenient; it is essential. A sustainable chemical industry requires rapid innovation in catalysis, separations, energy storage, recyclability, and green synthesis[10]. Manual experimentation simply cannot meet these demands at the necessary pace. In stark contrast, current degree programmes provide limited formal training in these areas, despite their growing importance in research and industry[11].

**The emerging professional profile of the chemist.** The required transformation fundamentally changes the skill set expected of future chemists. Data literacy, computational chemistry, process automation, statistical learning, and algorithmic reasoning are no longer optional. They are becoming core competencies. The competitive chemist of the future will operate at the interface of chemistry, materials science, chemical and mechanical engineering, computer science, and environmental science. Robotics competency, including installation, calibration, and operation of automated synthesis platforms, is increasingly vital. So too is familiarity with digital laboratory infrastructure, laboratory information management systems, data handling principles, and standardized automation protocols. Sustainability adds another dimension. Chemists must increasingly understand life-cycle analysis, carbon accounting, circular economy models, regulatory frameworks, and systems thinking, areas once considered peripheral to chemistry education.

**Why universities must reinvent their curricula.** The rise of automation and AI means that teaching chemistry in an apprentice-style, synthesis-focused training manner, as if it were still the late 20th century, is untenable. AI-driven algorithms propose synthetic routes, predict properties, and design molecules faster than any human could[12]. Automated instrumentation performs thousands of experiments per week with a level of reproducibility that manual work cannot match. Tomorrow's chemists, therefore, need to be comfortable in both environments, the wet lab and the digital lab. They must combine molecular intuition with computational thinking.

Most chemistry degree programmes still treat computational chemistry, despite its recognition in the 2013 Nobel Prize in Chemistry[13] awarded to Martin Karplus, Michael Levitt, and Arieh Warshel, as a niche elective; data literacy receives only minimal attention, and sustainability remains frequently peripheral. Cross-faculty teaching remains an illusion. Therefore, Universities must deeply rethink their structures; merely rebranding course programmes, making cosmetic updates, or adjusting academic credit points or course catalogues will not suffice. A meaningful transformation requires at least three structural changes.

- Chemistry needs a fundamental paradigm shift. Hands-on synthetic training will remain essential for conceptual understanding, but must be complemented by digital and automated experimentation. Students should learn to operate robotic platforms as naturally as they learn to perform titrations.
- Programming, statistics, AI, and laboratory automation should be core competencies, not add-ons or specialist tracks.
- This transformation must be strategic, not incremental. Artificial departmental boundaries - chemistry, computer science, mechanical engineering, and electrical engineering - must be overcome.

Successful digital chemistry programmes will require integrated curricula that transcend these administrative silos. This change will require institutional courage: shared teaching responsibilities, cross-faculty laboratories, co-developed degree programmes, and a redefinition of what constitutes *chemistry education*.

One pragmatic model for implementation is the introduction of parallel training pathways within chemistry programmes. Alongside a traditional curriculum, departments could develop a digital chemistry track integrating AI, automation, and robotics. Both pathways would share a common foundation in chemistry but diverge in emphasis, allowing students to specialize without requiring immediate, large-scale curriculum restructuring. Such a dual-track programme can function as an experimental laboratory for curriculum development, enabling departments to pilot new teaching formats, integrate emerging technologies, and iteratively evolve programme structures. Over time, such hybrid models may converge into fully integrated curricula. Teaching should be organized on a cross-faculty basis to provide students with integrated, interdisciplinary learning environments that reflect real-world problem-solving structures and better prepare them for complex professional contexts.

**A transformation that institutions ignore at their peril.** The transformation, however, is uneven. While Ph.D. programs have already begun shifting toward hybrid expertise, undergraduate education must follow soon if the gap between industrial practice and academic training is not to widen further. Many Universities still rely on curricula that underestimate how rapidly chemical research is changing. Programmes that fail to modernize risk preparing students for a world that no longer exists, while losing young talent to more agile fields. The declining popularity of chemistry is not evidence that chemistry is irrelevant; it is evidence that its educational structures no longer reflect the realities of the discipline. A 2015 study demonstrates that the portrayal of chemistry in early education diverges markedly from the realities of professional chemical practice[14].

The increasingly market-driven logic of higher education has created a situation in which degrees are treated much like financial assets. It is, in many respects, akin to the stock market: education has become a commodity whose perceived value fluctuates with interests, expectations, labour-market trends, and career opportunities. Students, acting as rational investors, select programmes based on anticipated returns, while Universities compete for *customers* through branding, rankings, and performance metrics. This dynamic introduces volatility, as fields associated with higher perceived returns, career prospects, job security, or societal prestige surge in enrolment, while disciplines essential to long-term societal needs, such as chemistry, may decline when their immediate *market price* is perceived as unfavourable. The metaphor underscores a deeper concern: as

education becomes commodified, its intrinsic scholarly, cultural, and strategic value risks being overshadowed by market behaviour that is often shortsighted and misaligned with the scientific challenges society urgently needs to address.

In this market-oriented education system, Universities and their administrations themselves participate actively in the commodification of degrees. When academic programmes are treated like tradable goods, institutions respond as actors in a competitive marketplace: through rankings, excellence initiatives, accreditation labels, targeted advertising, social-media campaigns, and branding strategies. Degree programmes are positioned like products promising employability, salary prospects, international networks, or fast-track careers. The success of a department is increasingly measured by metrics such as third-party funding, publication indices, or its standing in the Shanghai or Times rankings. This logic reinforces short-term trends: fields with a strong market image thrive, while disciplines with more complex narratives, such as chemistry, lose visibility and appeal. Universities thus often, albeit unintentionally, amplify a system in which students choose degrees based on perceived return on investment rather than scientific curiosity or societal relevance. As Harry Kroto noted: *the present market-driven complacency over maintaining the flow of scientists, in particular chemists, is a recipe for disaster*[1].

Paradoxically, chemistry is more essential than ever. The global transition to sustainable energy, recyclable materials, green manufacturing, and circular chemical processes depends critically on chemists with expertise in automation, AI, and systems design[15]. The central question is no longer whether chemistry is relevant, but whether Universities will adapt quickly enough.

**A discipline evolving, not declining.** The narrative that chemistry is losing value because enrolments are falling is profoundly misleading. What we are witnessing is a structural transformation, arguably the most significant since the emergence of modern physical chemistry. Universities, degree programmes, and students who embrace the new paradigm will find themselves uniquely positioned to shape the industries of the future: energy, electronics, materials, health, and sustainability.

In this new landscape, chemistry is not only the science of molecules; it is the science of systems[16]. It integrates AI-driven discovery, computation, automation, and global environmental constraints. Its value is expanding, not diminishing. Universities that fail to integrate engineering, computer science, and chemistry into modernized digital chemistry programmes risk stagnation. Institutions that succeed will train the innovators who redefine what chemical research and industry can achieve. The chemist of the future will be a hybrid professional:

- Part scientist, mastering molecular design and chemical reactivity
- Part engineer, fluent in instrumentation, automation, and reactor design
- Part data scientist, capable of modeling, prediction, and algorithmic thinking
- Part sustainability expert, aware of environmental and societal constraints

Such professionals will be comfortable in the laboratory, adept at coding, and able to navigate interdisciplinary collaborations with ease.

**In short: chemistry is not dying - it is evolving.** Last but not least, teaching programmes in digital chemistry bring automation and robotics to University research labs, thus offsetting declining student numbers while attracting a broader and more diverse cohort, including students with interests in computer science, robotics, and engineering. In the long run, such interdisciplinary curricula may not only increase enrolment but also strengthen institutional competitiveness and visibility, as areas such as AI-driven discovery and automated experimentation are strategically prioritized in funding landscapes. At the same time, shared cross-faculty teaching and research platforms enable more efficient use of infrastructure and foster collaboration across disciplines. In addition, these programmes align training more closely with industry needs, improving graduate employability and expanding career pathways. By positioning themselves at the forefront of digital and sustainable chemistry, Universities can reinforce their societal relevance in an increasingly competitive academic environment and contribute directly to future research and industrial innovation.

Yes, it is still worthwhile studying chemistry, because in a world defined by energy, materials, sustainability, and AI-driven innovation, chemistry is not fading but becoming the discipline on which all others depend. Those who succeed in this transformation will be the next global leaders in chemistry.

**Dirk G. Kurth** [ORCID] ✉

Chemische Technologie der Materialsynthese Fakultät für Chemie und Pharmazie Julius-Maximilians-Universität Würzburg Röntgenring, Würzburg, Germany. ✉e-mail: dirk.kurth@uni-wuerzburg.de

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

### Funding

### Competing interests
The author declares no competing interests.
