## [Transparent Peer Review file · Communications Chemistry]

Reframing Chemistry Education in the Age of Automation and AI

Corresponding Author: Professor Dirk Kurth

Version 0:

Reviewer comments:

Reviewer #1

(Remarks to the Author)

The authors of this letter argue that the changes to the broader landscape of chemists in the workplace, regardless of where that workplace is located, necessitate commensurate changes to the practice of teaching chemistry. In particular, the authors argue that the "rise of AI and automation" and the increasing utility of computational chemistry require three main changes to the broader chemistry curriculum. They state that students should be instructed in automated and digital experimentation alongside traditional synthetic methods, that skills such as programming, machine learning, automation, and probability/statistics should be core competencies, and that there should be more cross-departmental educational opportunities with departments such as computer science, mechanical engineering, and others.

The author's treatment of a degree as holding a market value, while disappointing to acknowledge, is ultimately a clear-eyed observation of the current state of academia. Further, there are strong arguments made in this letter as to the utility and essential nature of chemistry that I find quite compelling.

However, there is also material that I find to be extraneous and only serves to doom-say about the state of chemists in industry which the authors never propose solutions for. In particular, the second paragraph of the "Decline in Enrollments - a Shift in Perception" is entirely about student perceptions of chemical industry stability, something that changing an undergraduate curriculum is wholly unsuited to changing, and the entire "Rising Competition from Other Fields" section, that ultimately points out that most chemistry graduates do not end up in the field.

My main concern with this letter is that it is heavy on a view from 30,000 feet and does not offer any explicit suggestions as to where to begin this transition for an interested department. Few departments are able to undergo a complete curriculum change at the drop of a hat, and I find it unlikely that the letter author's would suggest such a drastic shift. There is no first step proposed, whether that be the introduction of DFT calculations to an organic or inorganic class to better visualize molecular orbitals, the introduction of data-science heavy genomics or proteomics techniques to a biochemistry laboratory class, or the explicit exploration of systems level thinking and sustainability considerations in introductory chemistry classes. All of these are, in this reviewer's view, accessible for many institutions, allow for individual faculty to explore and implement these ideas without need for a complete departmental overhaul, and naturally lead to additional changes and revisions as time goes by.

I do certainly feel that the letter as written would influence thinking in the field, but I feel it would be even more impactful with the changes listed above being made. In particular, a narrower focus on the portions of the chemistry industry that current degree programs are unable to provide for and specific suggestions for the first steps in meeting that challenge.

Reviewer #2

(Remarks to the Author)

"Is It Still Worthwhile Studying Chemistry? – Chemical Education in the Age of Automation."

Overall, an interesting and thought-provoking commentary piece on the future of chemistry education, industry and related employment.

Specific Comments:

Title – “Chemistry Education” to match other instances in article? Although there is of course distinction between machine learning and AI, should AI be mentioned in the title, given that it is perceived to be a paradigm shift moment?

Page 1 – Define PFAS.

Page 2 – “soft skills” should be “transferable skills”. Soft skills has connotations of ‘easier’ than other parts of the subject, when these skills can actually be difficult for students to master, particularly when not explicitly featured (as noted in the article) and/or supported in the curriculum.

Page 2 – Although probably relatively straightforward to find, the 2024 Nobel Prizes in Chemistry and Physics could be supported by a references, give the significance?

Page 3 – “chemical education” can be changed to “chemistry education” to match later discussion.

Page 3 – “apprenticeship-style” should be “apprentice-style” as the term ‘apprenticeship’ has specific qualification and regulatory framework meaning, for example in the UK.

Page 3 – Similar comment on reference to Nobel Prize, as above.

Page 3 – “elusion” should be “illusion”.

Page 4 – “programms” should be “programmes”.

Page 4 – Define ECTS.

Page 4 – Top section makes many valid points, but the structure could do with refinement to make it easier to read, as there are nested colons. Could be changed to “A meaningful transformation requires at least three structural changes.” full stop? Or a bullet point list, as used later in the article?

Page 5 – Top section makes important points, however the authors may wish to clarify “universities” as university management, as central university recruitment/marketing often governs the recruitment goals and marketing narrative taken, rather than particular subject academics such as in chemistry. Indeed, university senior leadership increasingly treat universities as businesses rather than an academic endeavour to educate and progress fields, such as novel research areas (or indeed whole subjects) being stopped as they are not achieving a target financial return. Clearly a university must be financially sustainable overall, but the closure of break-even or expensive subjects to teach, such as practical science, is both concerning and short-sighted.

Page 5 – “desaster” should be “disaster”.

Page 5 – “course prpgramms” should be “degree programmes”.

Page 6 – “programms” should be “programmes” in several occurrences, “enrollment” should be “enrolment”, “chemsity” should be “chemistry”, “interdisciplinay curriculum” should be “interdisciplinary curriculum”.

Page 6 – “University” capitalised, when not done elsewhere?

Page 6 – “The interdisciplinay curriculum may finally increase career opportunities beyond traditional chemistry employment.” Given that chemistry graduate destinations are already diverse and not restricted to traditional chemistry roles, this would be better as “The interdisciplinary curriculum may further increase career opportunities beyond traditional chemistry employment.”

Page 6 – “Yes, it is still worth while studying chemistry because in...” would be better as “Yes, it is still worthwhile studying chemistry, because in...”

Page 6 – Reference 2 missing closing bracket.

Version 1:

Reviewer comments:

Reviewer #1

(Remarks to the Author)

The authors have addressed my concerns more than satisfactorily. I find this work both relevant and convincing.

I thank the reviewers for their careful reading of the manuscript and for the constructive and thoughtful comments. I am encouraged that the reviewers find the central arguments compelling and appreciate the relevance of the topic. Below, we respond point-by-point.

Reviewer #1 (Remarks to the Author):

Comment: The authors of this letter argue that the changes to the broader landscape of chemists in the workplace, regardless of where that workplace is located, necessitate commensurate changes to the practice of teaching chemistry. In particular, the authors argue that the "rise of AI and automation" and the increasing utility of computational chemistry require three main changes to the broader chemistry curriculum. They state that students should be instructed in automated and digital experimentation alongside traditional synthetic methods, that skills such as programming, machine learning, automation, and probability/statistics should be core competencies, and that there should be more cross-departmental educational opportunities with departments such as computer science, mechanical engineering, and others.

The author's treatment of a degree as holding a market value, while disappointing to acknowledge, is ultimately a clear-eyed observation of the current state of academia. Further, there are strong arguments made in this letter as to the utility and essential nature of chemistry that I find quite compelling.

Answer: Thank you for this positive and encouraging comment. I appreciate the reviewer's recognition of both the realism of the argument and the strength of the case made for the continued importance of chemistry.

Comment: However, there is also material that I find to be extraneous and only serves to doom-say about the state of chemists in industry which the authors never propose solutions for. In particular, the second paragraph of the "Decline in Enrollments - a Shift in Perception" is entirely about student perceptions of chemical industry stability, something that changing an undergraduate curriculum is wholly unsuited to changing, and the entire "Rising Competition from Other Fields" section, that ultimately points out that most chemistry graduates do not end up in the field.

Answer: I respectfully disagree with this interpretation. In the present manuscript, the discussion of labour-market dynamics and the evolving role of chemists in industry is not intended as commentary on the state of the profession per se, but as a component of the argument. The manuscript frames student choice as being driven by the perceived alignment between education, scientific practice, and future career prospects.

Within this framework, the sections addressing industrial transformation and competition from other fields establish the decision context in which prospective students evaluate chemistry. These perceptions are not treated as external or incidental; rather, they are directly linked to educational structures, as stated explicitly: “students interpret the relevance of a discipline through its training structures and its visible connections to emerging technologies.”

Accordingly, the discussion of industry is not intended as “doom-saying,” but provides the causal link between perception and curriculum. The manuscript’s central claim is that when curricula fail to reflect contemporary developments, such as automation, AI-driven discovery, and interdisciplinary integration, the discipline is perceived as less future-oriented, irrespective of its actual importance. Thus, the criticised sections are essential in demonstrating that the issue is not a decline in chemistry’s relevance, but a misalignment between educational representation and disciplinary reality. If chemistry curricula do not reflect modern research practice, this perception gap is reinforced. This misalignment directly motivates the proposed structural changes in degree programmes.

Importantly, the manuscript does not seek to propose solutions to industrial challenges themselves. The argument is more specific: curricula do not alter economic conditions, but they do shape how the discipline is perceived by students. Rather, the manuscript addresses how educational systems respond to and signal these changes. For these reasons, the discussion of industry and perception is not extraneous but foundational to the manuscript’s argument, as it establishes why curricular transformation is both necessary and urgent.

Empirically, chemistry enrolments have not kept pace with growth in other fields, reflecting a shift in student preferences toward disciplines perceived as offering clearer career trajectories. In this sense, labour-market perceptions and competition from other fields are directly linked to the central argument of the manuscript: the need for curricular reform. For example, promotional material for chemistry programmes often emphasises traditional laboratory imagery (e.g., young people in lab coats and safety goggles in front of fume hoods), whereas competing disciplines foreground digital technologies, robotics, and computational environments, signals that may more strongly align with students’ expectations of future-oriented careers. The revised manuscript, therefore, strengthens the connection between these sections and the central thesis, while retaining their role in establishing the basis for the need for reform in chemistry education.

Comment: My main concern with this letter is that it is heavy on a view from 30,000 feet and does not offer any explicit suggestions as to where to begin this transition for an interested department. Few departments are able to undergo a complete curriculum change at the drop of a hat, and I find it unlikely that the letter author’s would

suggest such a drastic shift. There is no first step proposed, whether that be the introduction of DFT calculations to an organic or inorganic class to better visualize molecular orbitals, the introduction of data-science heavy genomics or proteomics techniques to a biochemistry laboratory class, or the explicit exploration of systems level thinking and sustainability considerations in introductory chemistry classes. All of these are, in this reviewer's view, accessible for many institutions, allow for individual faculty to explore and implement these ideas without need for a complete departmental overhaul, and naturally lead to additional changes and revisions as time goes by.

Answer: Thank you for this valuable comment. The revised manuscript proposes a pragmatic pathway based on a dual programme structure, in which departments retain a traditional curriculum while simultaneously developing a (hopefully cross-faculty) digital chemistry track. Both pathways share a common foundation in chemistry but diverge in emphasis, allowing incremental evolution without immediate, large-scale curriculum restructuring. Crucially, this dual-track model is not merely conceptual, but it can serve as an experimental "laboratory" for curriculum development, enabling departments to pilot new teaching formats, integrate emerging technologies, and iteratively evolve programme structures based on experience.

This approach directly addresses feasibility constraints raised by the reviewer: it provides a concrete starting point that can be implemented within existing institutional frameworks, while avoiding the need for abrupt, wholesale reform. At the same time, it preserves the manuscript's central argument that long-term convergence toward fully integrated, interdisciplinary curricula will be necessary.

In addition, the manuscript emphasises that teaching should be organised on a cross-faculty basis, ensuring that chemistry education reflects real-world problem-solving structures at the interface of disciplines. This further operationalises the proposed transformation by embedding it within existing institutional collaborations. Accordingly, the revised manuscript does not remain at a purely abstract level but offers a scalable, staged implementation strategy that aligns practical feasibility with structural change.

Comment: I do certainly feel that the letter as written would influence thinking in the field, but I feel it would be even more impactful with the changes listed above being made. In particular, a narrower focus on the portions of the chemistry industry that current degree programs are unable to provide for and specific suggestions for the first steps in meeting that challenge.

Answer: Thank you again for this helpful suggestion. Accordingly, the revised manuscript combines a clear strategic argument with a feasible entry point for implementation, thereby addressing both the conceptual and practical dimensions of the transformation. These changes aim to make the argument more focused and actionable while retaining the broader perspective of the manuscript.

Reviewer #2 (Remarks to the Author):

Comment: Overall, an interesting and thought-provoking commentary piece on the future of chemistry education, industry and related employment.

Answer: Thank you for this positive and encouraging assessment. I appreciate the reviewer's recognition of the manuscript as both interesting and thought-provoking. The intention of this Comment is precisely to stimulate discussion at the interface of chemistry education, industrial practice, and employment landscapes and to contribute to a broader re-evaluation of what constitutes chemistry education in the context of ongoing technological and societal transformation.

Specific Comments:

Comment: Title – “Chemistry Education” to match other instances in article? Although there is of course distinction between machine learning and AI, should AI be mentioned in the title, given that it is perceived to be a paradigm shift moment?

Answer: Changed to: “Is It Still Worthwhile Studying Chemistry? – Chemistry Education in the Age of Automation and AI.” To be more concise I use AI-driven discovery throughout the text now.

Page 1 – Define PFAS. Corrected

Page 2 – “soft skills” should be “transferable skills”. Soft skills has connotations of ‘easier’ than other parts of the subject, when these skills can actually be difficult for students to master, particularly when not explicitly featured (as noted in the article) and/or supported in the curriculum. Good point. Corrected.

Page 2 – Although probably relatively straightforward to find, the 2024 Nobel Prizes in Chemistry and Physics could be supported by a references, give the significance?

I added two references (4, 5)

Page 3 – “chemical education” can be changed to “chemistry education” to match later discussion. Corrected.

Page 3 – “apprenticeship-style” should be “apprentice-style” as the term ‘apprenticeship’ has specific qualification and regulatory framework meaning, for example in the UK. Corrected.

Page 3 – Similar comment on reference to Nobel Prize, as above.

I added a references (13)

Page 3 – “elusion” should be “illusion”. Corrected.

Page 4 – “programmms” should be “programmes”. Corrected.

Page 4 – Define ECTS. Changed to academic credit points.

Page 4 – Top section makes many valid points, but the structure could do with refinement to make it easier to read, as there are nested colons. Could be changed to “A meaningful transformation requires at least three structural changes.” full stop? Or a bullet point list, as used later in the article? Corrected.

Page 5 – Top section makes important points, however the authors may wish to clarify “universities” as university management, as central university recruitment/marketing often governs the recruitment goals and marketing narrative taken, rather than particular subject academics such as in chemistry. Indeed, university senior leadership increasingly treat universities as businesses rather than an academic endeavour to educate and progress fields, such as novel research areas (or indeed whole subjects) being stopped as they are not achieving a target financial return. Clearly a university must be financially sustainable overall, but the closure of break-even or expensive subjects to teach, such as practical science, is both concerning and short-sighted.

In my experience, both university administrations and academic staff can be reluctant to implement substantial changes to degree programmes. My perspective is primarily shaped by the German system, where universities are predominantly public institutions and administrative structures are, to a significant extent, embedded within the academic staff itself. I am, however, aware that in other systems, universities are often governed by professional management structures, where recruitment strategies and programmatic decisions are more centrally coordinated. So, I changed the text to: ‘Universities and their administrations’.

Page 5 – “desaster” should be “disaster”. Corrected.

Page 5 – “course prpgramms” should be “degree programmes”. Corrected.

Page 6 – “programmms” should be “programmes” in several occurrences, “enrollment” should be “enrolment”, “chemsity” should be “chemistry”, “interdisciplinay curriculum” should be “interdisciplinary curriculum”. Corrected.

Page 6 – “University” capitalised, when not done elsewhere? Corrected.

Page 6 – “The interdisciplinary curriculum may finally increase career opportunities beyond traditional chemistry employment.” Given that chemistry graduate destinations are already diverse and not restricted to traditional chemistry roles, this would be better as “The interdisciplinary curriculum may further increase career opportunities beyond traditional chemistry employment.” Corrected.

Page 6 – “Yes, it is still worth while studying chemistry because in...” would be better as “Yes, it is still worthwhile studying chemistry, because in...” Corrected.

Page 6 – Reference 2 missing closing bracket. Corrected.

Finally: Additional references are highlighted in the manuscript.